# A Degradable Inverse Vulcanized Copolymer as a Coating Material for Urea Produced under Optimized Conditions

**DOI:** 10.3390/polym13224040

**Published:** 2021-11-22

**Authors:** Ali Shaan Manzoor Ghumman, Rashid Shamsuddin, Mohamed Mahmoud Nasef, Efrem G. Krivoborodov, Sohaira Ahmad, Alexey A. Zanin, Yaroslav O. Mezhuev, Amin Abbasi

**Affiliations:** 1Chemical Engineering Department, Universiti Teknologi PETRONAS, Bandar Seri Iskandar 32610, Perak Darul Ridzuan, Malaysia; ali_19001079@utp.edu.my (A.S.M.G.); amin_18000407@utp.edu.my (A.A.); 2HICoE, Centre for Biofuel and Biochemical Research (CBBR), Institute of Sustainable Building, Universiti Teknologi PETRONAS, Seri Iskandar 32610, Perak, Malaysia; 3Department of Chemical and Environmental Engineering, Malaysia Japan International Institute of Technology, Universiti Teknologi Malaysia, Jalan Sultan Yahya Petra, Kuala Lumpur 54100, Malaysia; mahmoudeithar@cheme.utm.my; 4Institute of Chemistry and Sustainable Development, Dmitry Mendeleev University of Chemical Technology of Russia, Miusskaya Sq. 9, 125047 Moscow, Russia; vv1992@yandex.ru (E.G.K.); zanin@muctr.ru (A.A.Z.); valsorja@mail.ru (Y.O.M.); 5Department of Electrical Engineering, Wah Engineering College, University of Wah, Wah Cantt 47040, Punjab, Pakistan; sohaira.ahmad@wecuw.edu.pk

**Keywords:** slow-release fertilizers, inverse vulcanization, urea, dip-coating, sulfur polymers, response surface methodology, central composite design, coated urea

## Abstract

Global enhancement of crop yield is achieved using chemical fertilizers; however, agro-economy is affected due to poor nutrient uptake efficacy (NUE), which also causes environmental pollution. Encapsulating urea granules with hydrophobic material can be one solution. Additionally, the inverse vulcanized copolymer obtained from vegetable oils are a new class of green sulfur-enriched polymer with good biodegradation and better sulfur oxidation potential, but they possess unreacted sulfur, which leads to void generations. In this study, inverse vulcanization reaction conditions to minimize the amount of unreacted sulfur through response surface methodology (RSM) is optimized. The copolymer obtained was then characterized using Fourier transform infrared spectroscopy (FTIR), thermogravimetric analysis (TGA), and differential scanning calorimetry (DSC). FTIR confirmed the formation of the copolymer, TGA demonstrated that copolymer is thermally stable up to 200 °C temperature, and DSC revealed the sulfur conversion of 82.2% (predicted conversion of 82.37%), which shows the goodness of the model developed to predict the sulfur conversion. To further maximize the sulfur conversion, 5 wt% diisopropenyl benzene (DIB) as a crosslinker is added during synthesis to produce terpolymer. The urea granule is then coated using terpolymer, and the nutrient release longevity of the coated urea is tested in distilled water, which revealed that only 65% of its total nutrient is released after 40 days of incubation. The soil burial of the terpolymer demonstrated its biodegradability, as 26% weight loss happens in 52 days of incubation. Thus, inverse vulcanized terpolymer as a coating material for urea demonstrated far better nutrient release longevity compared with other biopolymers with improved biodegradation; moreover, these copolymers also have potential to improve sulfur oxidation.

## 1. Introduction

The global population, which is 7.9 billion today [1,2], will exponentially grow to 10 billion by 2050. Hence, for the survival of humanity and for food security, enhancing crop production is needed while reducing the environmental population, and preserving soil health will be a challenge. To boost crop yields, the agricultural sector is posing a solution to consume huge amounts of nitrogen fertilizers, which altogether add up to adverse consequences [3,4,5]. Urea is the most essential nitrogen fertilizer; however, it is vulnerable to losses due to surface run-off, leaching, and ammonia volatilization, thus disturbing the neighbor ecosystem [6,7]. It has been estimated that almost 70% of the total urea applied to the crops dissipate to the environment causing low nutrient use efficiency (NUE) and high production cost [8,9,10].

To cease the mounting problem and achieve agronomic and environmental benefits, agricultural researchers and industries have been working to develop novel slow-release fertilizers (SRFs). Slow-release fertilizers are deliberately fabricated manure that delayed the release of the nutrient in synchrony with the nutrient requirement of the crops, hence, increasing the crop yield and nutrient uptake efficacy (NUE) [11]. To date, various materials have been utilized to develop SRF, which includes synthetic and natural polymers and inorganic materials. Although synthetic polymers have demonstrated promising results in terms of nutrient release longevity, the involvement of harmful solvent in the coating of synthetic polymers on urea and non-biodegradability leads to environmental and soil pollution [12,13,14,15]. On the other hand, natural polymers are suffering from their hydrophilic nature, which leads to the abrupt release of the nutrients at an unpredictable time [5]. The brittle nature of the inorganic material such as sulfur promotes the generation of micropores on the coating surface, causing failure in halting the nutrient release [5,16,17,18]. Such formidable factors arise a need to look for other coating materials that are green, sustainable, and have better physicochemical properties.

Sulfur polymers are a new class of green and sustainable polymers produced via a newly developed method called inverse vulcanization. It is a green polymerization process since it does not require any initiators or solvents and also due to the fact that it is highly atom-economical [19,20]. Further, it utilizes the already available and cheap elemental sulfur as the main comonomer, which is openly piled up as a byproduct in gas and petroleum refineries, causing many environmental problems [21,22]. Inverse vulcanization was first reported in 2013 by Pyun et al. as a polymerization technique that uses the same principles of rubber vulcanization; however, in this case, sulfur plays as the main comonomer [19,23]. Three different classes of comonomers, i.e., petro-based, bio-based, and vegetable oils, are utilized in the production of sulfur-based polymers. Vegetable oils consist of an unsaturated portion and a saturated portion, of which the unsaturated portion can act as a comonomer to produce sulfur-based polymers; nevertheless, the complex structure of vegetable oils and also their impurity (saturated portion) make it more difficult to produce controlled sulfur-based polymers using vegetable oils as monomers [21,23,24]. Oils of different vegetables including canola [25,26,27,28], castor [29], rubber seed [30,31], palm [32], linseed [33], corn [34], olive [33], sunflower [33], rice bran [29], soybean [35], and cottonseed [36] have been employed as monomers in the production of sulfur-enriched polymers. Due to the presence of the unsaturated section of vegetable oils, their copolymerization with sulfur results in composite structures because of the presence of the unreacted sulfur. The morphological properties of these composites are highly dependent on the composition of the utilized vegetable oil [32,34]. These polymers have been investigated in several applications, such as Li-S battery cathodes, mercury removal, hydrocarbon removal, and fertilizers [21].

Despite the fact that vegetable-oil-based copolymers have demonstrated promising results in many applications, they are still suffering from some challenges. For example, the presence of unreacted sulfur adversely affects their performance in Li-S batteries as it contributes to the capacity fading of the battery [24].

Sulfur is a secondary, yet indispensable nutrient required for plant growth; Stella F. Valle et al. reported that inverse vulcanized copolymer has the potential to improve sulfur oxidation, hence, providing SO_4_^2−^ in a more convenient way compared with elemental sulfur [35]. The high sulfur content, better sulfur oxidation, and the biodegradable nature of these copolymers have attracted and planted a seed for this research. However, the presence of unreacted sulfur particles can promote the generation of the micropores on the surface of the copolymers, which could cause the sudden release of the nutrient. In lab investigation, it is observed that by optimizing the reaction conditions, the amount of unreacted sulfur can be controlled.

Herein, the synthesis of the inverse vulcanized copolymer under optimized conditions is reported in this study. The reaction conditions are optimized using response RSM through central composite design (CCD). The produced copolymer is then characterized using Fourier transform infrared spectroscopy (FTIR), thermogravimetric analysis (TGA), and Differential scanning calorimetry (DSC). Terpolymer is produced to further reduce the amount of unreacted sulfur and utilized to coat the urea to produce slow-release fertilizer (SRF). The morphology of the coated urea is studied using scanning electron microscopy (SEM), and their nutrient-release longevity is investigated in distilled water. A soil burial test is conducted to assess the biodegradability of the copolymer. A schematic figure representing the research work is given in Figure 1.

## 2. Materials and Methods

### 2.1. Materials

Elemental Sulfur (reagent grade) and jatropha oil (JO) were purchased from PC laboratory reagents, Malaysia, and Kinetics Chemicals Sdn Bhd, Malaysia, respectively. Diisopropenyl benzene, diacetyl monoxime, thiosemicarbazide (TSC), Phosphoric acid, sulfuric acid, and tetrahydrofuran were purchased from Sigma-Aldrich. Urea (AR-grade) was procured from PETRONAS Fertilizer Kedah Sdn Bhd, Malaysia. All materials were used as received without further purification.

### 2.2. Methods

#### 2.2.1. Optimization of Inverse Vulcanization Reaction Conditions

##### Design of Experiment

The design of the experiment was carried out using Design Expert Software (Version 12.0.12.0, from Stat-Ease, MN 55413, USA) for the optimization of the synthesis of the inverse vulcanized copolymers using response RSM with full factorial CCD. This type of design involves a two-factorial design (+1, −1) overlaid by the central points (0) and the star points (+α, −α) at the distance of α = 1.682 from the design center at the axis of each design variable.

Initial sulfur composition, reaction temperature, and reaction time are three selected independent variables for the optimization of the reaction condition to maximize the sulfur conversion in the final structure of the copolymer. Preliminary experiments were carried out to set the range of these independent variables by monitoring the resultant single phase of the copolymer and the release of hydrogen sulfide (H_2_S) gas, which directly affects the structure of the copolymer. Ranges of these factors, along with their levels, are presented in Table 1. As an example, the reaction between sulfur and Jatropha oil (JO) below 170 °C results in a two-phase product that indicates the incomplete reaction, while reaction at a temperature above 185 °C promotes the release of H_2_S gas, which results in the generation of the porous structured copolymer. The release of H_2_S gas was observed by blackening of the lead acetate solution wetted filter paper.

The response of the experiments—which is the conversion of the elemental sulfur to the polymeric sulfur chain—was calculated using the DSC thermogram of the resulting copolymers. The thermogram of the elemental sulfur shows the endotherms, which represent the phase transitions of the elemental sulfur from 102 to 120 °C; these are highly dependent on the weight of sulfur, as the linear integration of these endotherms shows.

The DSC thermogram of the inverse vulcanized copolymer also shows endotherm in these ranges, which represents the presence of the unreacted sulfur in the copolymer. As the intensity of these endotherms increases with the increase in sulfur weight, we ran a DSC analysis of sulfur with different weights and made a calibration graph to obtain the equation. To calculate the conversion of the sulfur, linear integration was carried out on the endotherms of the copolymers that appeared in DSC thermograms ranging from 102 to 120 °C and compared with the data obtained through the graphical equation. The linear integration was carried out with the help of TA instruments software.

##### Regression Model

The data obtained through CCD was analyzed using response surface regression and observed to best fit the quadratic model given in Equation (1). The statical procedures were followed to analyze the goodness-of-fit and the significance of the parameters of the regression model.
(1)$$Y=b_{o}+\sum_{i=1}^{n}b_{i}X_{i}+\sum_{i=1}^{n}b_{ii}X_{i}^{2}+\sum_{i=1}^{n}\sum_{j\geq 1}^{n}b_{ij}X_{i}X_{j},$$
where Y is the conversion (%) of the sulfur; *b_o_, b_i_, b_ii_*, and *b_ij_* are the constant, linear, squared, and interaction effect coefficients, respectively; and *X_i_* and *X_j_* are the coded values of the variables *i* and *j,* respectively.

#### 2.2.2. Synthesis of Copolymer

A 25-mL glass vial was filled with the designed weight of the elemental sulfur and placed in a thermoset oil bath preheated to a required reaction temperature under vigorous stirring to initiate the formation of the thiyl radicals. First, the elemental sulfur upon heating starts to melt, after which when the temperature reaches >159 °C, octet structure of the sulfur starts to open to form the thiyl radicals, which is accompanied by the color change from yellow color to orange color liquid; at this point, the designed amount of the jatropha oil is added in a dropwise manner to avoid a sudden decrease in temperature [30,31,32]. After adding jatropha oil to the glass vial, a plaque mixture was formed, which was allowed to react under vigorous stirring for the designed time. The design time, temperature, and sulfur/jatropha oil amount refer to the amount that is required to run the experiment as designed for the optimization of the reaction conditions.

The design of experiments is presented in Table 2, which is the combination of 2^3^ factorial points, 10 central points, and 2 axial points, summing up to 20 combinations. After the reaction mixture was allowed to react for the desired time, the glass vial was removed from the thermoset oil bath and placed under a fume hood to allow the product to cool at room temperature. During the reaction, it is highly recommended to carry the reaction under a fume hood because it may release toxic gas such as H_2_S.

#### 2.2.3. Copolymer Characterization

##### Fourier Transform Infrared Spectroscopy (FTIR)

FTIR analysis of the copolymer produced under optimized conditions was carried out to investigate the chemical composition and confirm the successful reaction of the thiyl radicals with the unsaturated part of the jatropha oil. The scan frequency range is between 500–4000 cm^−1^ with 4 cm^−1^ resolution. A total 8 number of scans were performed to confirm the chemical structure using the PerkinElmer frontier model spectrometer (PerkinElmer, Waltham, MA, USA). The attenuated total reflectance (ATR) method was used.

##### Thermogravimetric Analysis (TGA)

Thermal stability of the produced copolymer was evaluated at a temperature range of 25–800 °C with 10 °C heating rate using a PerkinElmer STA 6000 simultaneous thermal analyzer (PerkinElmer, Waltham, MA, USA) under nitrogen atmosphere.

##### Differential Scanning Calorimetry (DSC)

To evaluate the thermal properties and estimate the unreacted sulfur in the produced copolymer, a TA instruments Q2000 thermal analyzer (TA instruments, 159 Lukens Dr, New Castle, DE, USA) was used to obtain the DSC thermogram. Properties of the copolymer were evaluated with 20 °C/min heating rate at −80 to 200 °C temperature range under nitrogen atmosphere.

#### 2.2.4. Synthesis of Terpolymer

To further reduce the amount of the unreacted sulfur in the final copolymer, 5 wt% diisopropenyl benzene was used as a crosslinker. The terpolymer was synthesized using the same procedure as explained in Section 2.2.2 and using the optimized conditions.

#### 2.2.5. Coating of the Urea

To coat the urea granules, terpolymer was dissolved in tetrahydrofuran (THF) solvent to produce a coating solution, which was followed by a coating of the urea using the dip-coating method. The coating solution was prepared by dissolving 5 g of terpolymer in 6 mL THF and left overnight in an incubator shaker to make a homogenous mixer. After mixing, 10 g of urea with a size range of 2 to 2.5 mm was added in a polymer solution and gently stirred using a glass rod to obtain a uniform coating on the urea, followed by drying in an oven at 60 °C for 24 h.

#### 2.2.6. Morphology of the Coated Urea

Morphology of the coated urea was studied by scanning electron microscopy (SEM) using Zeiss EVO LS 15 microscope armed with Oxford Instruments INCAx-act EDX spectroscope (Carl Zeiss, Göschwitzer, Jena, Germany). To obtain the cross-section of the coated urea and to estimate the thickness of the coating, coated urea was cut in half using a sharp knife and coated with gold using a sputter coater (Emitech K550X) for SEM analysis.

#### 2.2.7. Nitrogen Release in Distilled Water

Total nitrogen content of the coated urea was estimated using the Kjeldahl method [37] before leaching test. After this, 2.0 g coated urea was placed in an Erlenmeyer flask filled with 200 mL of distilled water and sealed with clinging wrap to avoid water loss through evaporation. To measure the leached amount of the nitrogen into water after every 24 h, 2.5 mL of the gently stirred aliquot was taken out, and the water was replaced with 200 mL of fresh distilled water. The concentration of the urea in aliquot was found using diacetyl monoxime (DAM) calorimetry method, which uses red color solution. To obtain the red color solution, the aliquot was combined with 7.5 mL of the color reagent in a 60 mL glass vial and placed in a water bath at 85 °C for 30 min. The amount of urea in the sample determines the intensity of the color. The glass vials containing the solution were then put in ordinary tap water at a temperature of 20 °C for 20 min to cool down. The total release time was determined by doing triplicates and using the standard curve technique.

To make the DAM solution, 2.5 g of DAM was dissolved in 100 mL of distilled water; to make the TSC (Thiosemicarbazide) solution, 0.25 g of TSC was dissolved in 100 mL of distilled water; to make the acid reagent, 250 mL of phosphoric acid was combined with 240 mL of distilled water and 10 mL of sulfuric acid. Consequently, the colored reagent solution was made by carefully mixing 25 mL of DAM solution, 15 mL of TSC solution, and 460 mL of acid reagent.

#### 2.2.8. Soil Burial Test

A soil (sand 20.5%, silt 39.3% and clay 40.2%) burial test was conducted to investigate the biodegradability of the copolymer. For this purpose, 2 g of the copolymer was enclosed in a woven mesh bag (similar to a teabag) and buried under a soil in a polymer container. The bag was buried under the soil at a depth of 10 cm, and the soil was kept moist throughout the experiment. After a regular interval, the buried bag was taken out, washed with distilled water to remove the soil attached to it, dried in an oven to obtain a constant weight, and the weight loss of the copolymer was recorded using the method in [38].

## 3. Results and Discussions

### 3.1. Optimization of Inverse Vulcanization Conditions

#### 3.1.1. Design of Experiment and Regression Modeling

Design expert software (12.0.12.0) was used to design the experiment for optimization of the reaction conditions through response RSM using CCD. The design of the experiment, along with the responses, are given in Table 2 (Section 2.2.2). Analysis of variance (ANOVA) is crucial in determining the adequacy of the models; thus, ANOVA was used to analyze the fitness of all regression models, which revealed the highest validity of the quadratic model. No transformation of the data is required as the ratio of the minimum and the maximum response is 3.58 (81.06/22.62), which is less than 10. Fisher F test is conducted on the quadratic model and demonstrated its low sequential *p*-value (<0.0001) and high square of correlation value (R^2^ = 0.9838, Adjusted R^2^ = 0.9692 and predicted R^2^ = 0.9158) [39], indicating the significance of the model. Full ANOVA of the quadratic model is presented in Table 3. The signal-to-noise ratio (which is required to be greater than 4) is found to be 28.5325, revealing the adequate precision of the model [40], which indicates that this model can be used to navigate the design space.

There are only 0.01% chances that 67.44 F-value of the model occurs due to noise, indicating the significance of the model. The significance of terms of the model is demonstrated by *p*-value, which should be less than 0.05, but Table 3 shows some terms such as AC, BC, and B^2^, which mean that model reduction is required. The significant terms such as A, B, and C showed that selection of the parameters for optimization is appropriate, as the ANOVA revealed their significance in influencing the sulfur conversion. The model is reduced by ignoring the insignificant terms, and ANOVA for the reduced model is shown in Table 4. The reduced model has a high F-value of 119.01 with a low *p*-value of <0.0001 and high square correlation (R^2^ = 0.9821, Adjusted R^2^ = 0.9739 and predicted R^2^ = 0.9518), indicating that the significance of the model increased by ignoring the insignificant terms.

After removing the insignificant terms from the quadratic model, the final equation in terms of actual factors to predict the response is shown below as Equation (2)
(2)$$Y=60.77-17.48*A+2.61*B+4.77*C+2.58*AB-3.80*A^{2},$$
where *Y* is the conversion% of the sulfur to poly sulfur, *A* is the initial sulfur content (wt%), *B* is reaction temperature (°C), and *C* is the reaction time (min).

Figure 2a shows the normal dispersion of the error, indicating the adequacy of the model to predict the response in the experimental range. Figure 2b also demonstrates the good fitness of the model as the points of the graph between the actual and predicted response cluster around the straight line [39,40,41].

#### 3.1.2. Optimization of the Reaction Conditions

Figure 3 depicts the effect of the reaction temperature and initial sulfur loading at a different time on sulfur conversion. As can be seen, increasing the reaction temperature increases the sulfur conversion; however, the sulfur loading increases the amount of unreacted sulfur. Reaction time also has a positive impact on the sulfur conversion. With lower reaction time, the max conversion that can be achieved is ≥71%; increasing the time increases the conversion to ≤80%. To optimize these condition, constraints are set to keep all the reaction parameters within the limit and maximize the sulfur conversion in Design-Expert software. As a result, the software suggested 100 solutions, and we chose the best one that has high sulfur conversion. It is found that 82.37% conversion of sulfur can be achieved if 51.94 wt% S is allowed to react with jatropha oil for 74.21 min at 169.9 °C temperature.

### 3.2. Characterization of the Copolymer Produced under Optimized Conditions

Fourier Transform Infrared Spectroscopy (FTIR)

The copolymer produced under optimized conditions is then analyzed using FTIR to confirm the formation of the copolymer. The FTIR–ATR spectrum of copolymer and jatropha is depicted in Figure 4. The spectrum of jatropha contain cis-alkene character peaks at 1660 and 3009 cm^−1^ representing the stretching of C=C and C=C–H, which appeared due to the unsaturated part of jatropha oil [24,25,34]. However, these peaks disappear in the spectrum of the copolymer and a new peak appears at 804 cm^−1^, representing the vibration of C–H in vicinity of C–S bond, and thus, confirming the utilization of C=C to form C–S bond meaning a copolymer has been successfully formed [34,35].

Thermal stability of the copolymer is investigated using thermogravimetric analysis. TGA thermograms of copolymer, jatropha oil, and elemental are presented in Figure 5. Elemental starts to decompose at 200 °C and fully decomposes at 320 °C. Jatropha oil starts to degrade at a temperature of 289 °C in a two-step manner. The significant loss in one step is due the degradation of the polyunsaturated fatty acids followed by the decomposition of monounsaturated acids and remaining polyunsaturated acids, and it completely decomposes at 600 °C [42,43]. Meanwhile, the copolymer degrades in three steps; in the first step, loosely bonded and unreacted sulfur starts to degrade, which is onset at 205 °C, followed by the degradation of the oil part of the copolymer [24,29,34,44]. Copolymer yielded 18% char at 800 °C, which reflects its thermal stability. The copolymer is found to be thermally stable.

To evaluate the thermal properties and estimate the sulfur conversion, DSC analysis is carried out. DSC thermograms of the copolymer and elemental are shown in Figure 6. Two endotherms appeared in the thermogram of the sulfur at 103 and 119 °C, representing the crystalline nature of the sulfur [24,25,35]. However, only one endotherm was observed in the DSC thermogram of the copolymer formed, which represents the presence of unreacted sulfur in the copolymer. By carrying out integration of the copolymer endotherm, 17.8% unreacted sulfur is found. The obtained model predicted the conversion to be 82.37% while the actual conversion is 82.2%, which shows the goodness of the model.

### 3.3. Coating of Urea with Terpolymer

As revealed by DSC thermogram, there is still unreacted sulfur present in copolymer; to overcome this and minimize the unreacted sulfur, terpolymer were synthesized using the same optimized conditions except the 5% DIB was used as crosslinker, as explained in our early investigation reporting that addition of crosslinker reduces the unreacted sulfur [30,31]. After synthesis of terpolymer, urea granules were coated using the solution of terpolymer with THF using dip-coating method. The coated granules were then placed in oven for 24 h for drying.

Morphology of the obtained coated urea was then investigated by taking SEM images. Cross-section SEM images of the coated urea are shown in Figure 7, which clearly differentiate the urea (marked by yellow circle) and coating (marked by red circle). SEM images revealed the nonuniformity of the coating caused by the sticky nature of the copolymer, which promotes adhering of coated urea with each other. The SEM image also revealed that there is no unreacted sulfur present as no isolated particles appear on the surface of the copolymer. The thickness of the coating is found to be 206.31 µm.

### 3.4. Nitrogen Release in Distilled Water

Nitrogen release from coated urea is tested in distilled water using DAM calorimetry method. The nitrogen release profile of urea and the coated urea are shown in Figure 8. The initial nutrient release rate from the coated urea reflects the integrity of the coating; the stronger and more thorough the coating is, the slower the nitrogen release rate. The pristine urea releases almost 99.9% of its total nutrients within 24 h of incubation, whereas the coated urea delayed the release of the nutrients and released only 65% of its total nutrients after 40 days of incubation, which is far better compared with urea coated with biopolymer, which release nutrients in less than 5 days of incubation [38]. These promising results demonstrate the potential of these copolymers to be utilized as coating material for urea as they have shown results comparable to the synthetic-petroleum-based polymers.

Initial release rate of nutrient is very slow until the 10th day of incubation—this period is regarded as the lag period. This coated urea perfectly follows the European Standard (EN 13266, 2001) as it does not release 15% of nutrients in 24 h of incubation, which reflects the integrity of the coating film.

The release of urea is characterized by a tendency to auto acceleration (Figure 8), which is possibly associated with an increase in the cross-sectional area of the pores of the polymer shell with time. If we assume that the release of urea (nitrogen source) obeys first-order kinetics, which is characteristic of highly soluble substances, and the pores are closed with urea, which creates a temporary diffusion barrier, then Equation (3) can be written as
(3)$$-\frac{dN}{dt}=sk_{0}\left(1-N\right)$$
where N is the conversion of nitrogen (urea) release; s is the total cross-sectional area of the pores of the polymer shell; $k_{0}$ is the true constant of the rate of urea release; t is time.

As the pore-clogging urea dissolves, a linear increase in the total pore cross-sectional area of the polymer shell can be expected in accordance with Equation (4).
(4)s=αt
where α is the proportionality coefficient.

Substituting (4) into (3), dividing the variables, and integrating in the range from 0 to N and from 0 to t, we obtain Equation (5) as
(5)$$\ln\left(1-N\right)=kt^{2}$$
where $k=\frac{\alpha k_{0}}{2}$ is the rate-effective constant urea release.

Equation (5) is linear in the coordinates “$\ln\left(1-N\right)\;\mathrm{vs}.\;t^{2}$”, which allow determining the value of k from the tangent of the slope of the straight line (Figure 8a). The rate-effective constant urea release is 6.2 × 10^−4^ day^−2^, which allows calculating the theoretical kinetic release curve according to Equation (6) and comparing it with experimental data (Figure 8b).
(6)$$N=1-e^{kt^{2}}$$

As can be seen (Figure 8B), Equation (6) is in satisfactory agreement with the experimental data.

### 3.5. Soil Burial Test

Figure 9 shows the weight loss of the copolymer in soil; the weight loss of the copolymer increases with the increase in time of soil burial. The weight loss reaches 26% on the 52nd day of the incubation, demonstrating that the copolymer is degrading slowly in soil and will take longer to fully decompose. The degradation kinetics formally correspond to the zero-order equation with the rate constant 0.465% × day^−1^ (Figure 9).

The degradation starts with the sulfur oxidation from loosely bonded S–S and unreacted sulfur present in the copolymer as *A. niger* bacteria is present in the soil, which helps sulfur to oxidize. This test confirms the biodegradable nature of the copolymer, which is an additional benefit of using this material as coating material for urea, as this mitigates the problem of pollution caused by the coating shell left in the soil after release of nutrients. The degradation of these copolymers will also help the plant growth as the oxidation of the sulfur produces sulfate, which is an accessible form of secondary nutrient required by the plants. This test shows the biodegradable nature of the inverse vulcanized copolymers, which is degrading with extension of soil burial incubation period [35].

## 4. Conclusions

RSM was utilized to optimize the inverse vulcanization reaction condition to minimize the unreacted sulfur amount in the final copolymer. A quadratic model was developed to predict the sulfur conversion, and it was found that 82.37% conversion of sulfur can be achieved if 51.94 wt% S is allowed to react with jatropha oil for 74.21 min at 169.9 °C temperature. DSC revealed the actual conversion to be 82.2%, which shows the goodness of the developed model. To further maximize the sulfur conversion, 5 wt% DIB was used as a crosslinker, and the obtained terpolymer was utilized as a coating material to develop a novel slow-release coated urea to delay the nutrient release. The nutrient release test revealed that only 65% of the total nutrient released after 40 days of incubation compared with pristine urea, which released 99% in just one day. Biodegradability of terpolymer was revealed by soil incubation test, which showed that 26% weight loss occurred after 52 days of incubating the terpolymer in soil.

## Figures and Tables

**Figure 1 polymers-13-04040-f001:**
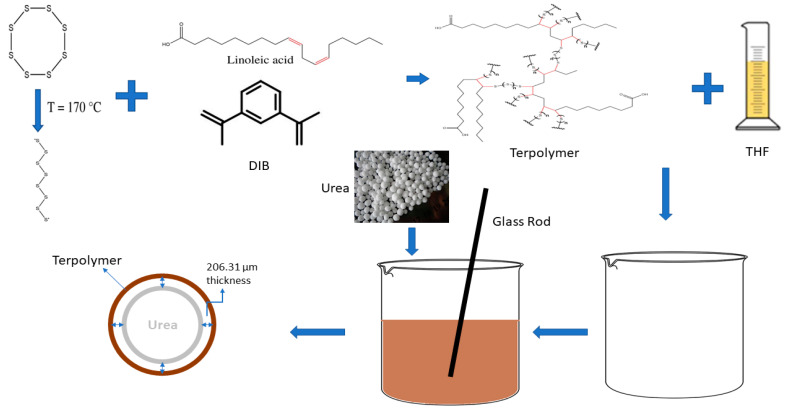
Schematic of the research work.

**Figure 2 polymers-13-04040-f002:**
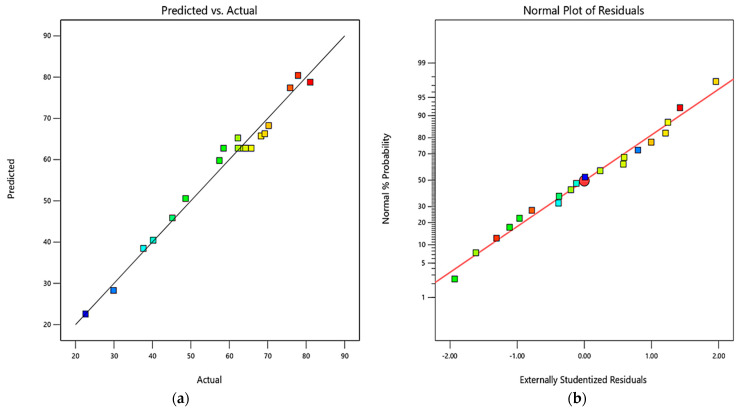
Plot of (**a**) Normal residuals and (**b**) Predicted vs. Actual Response.

**Figure 3 polymers-13-04040-f003:**
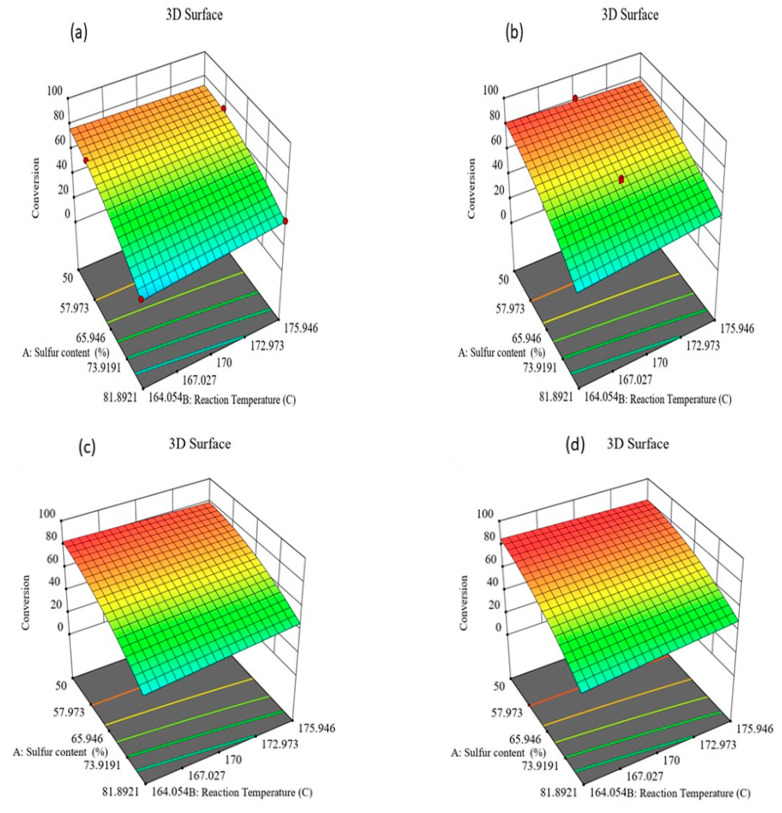
Influence of reaction conditions on sulfur conversion at different times (**a**) 42 min, (**b**) 60 min, (**c**) 65 min, and (**d**) 74.21 min.

**Figure 4 polymers-13-04040-f004:**
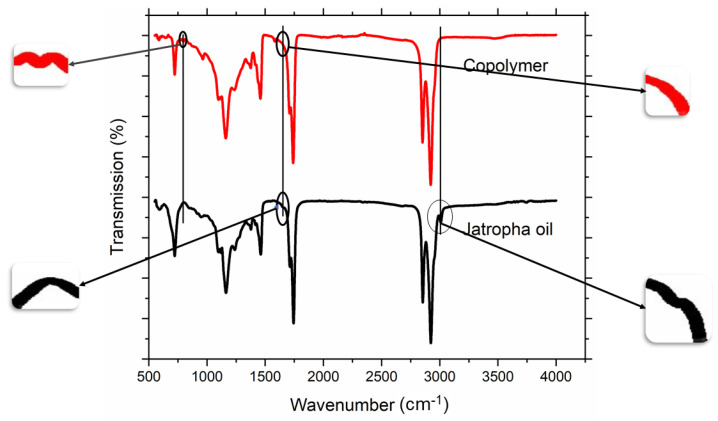
ATR–FTIR of the optimized copolymer.

**Figure 5 polymers-13-04040-f005:**
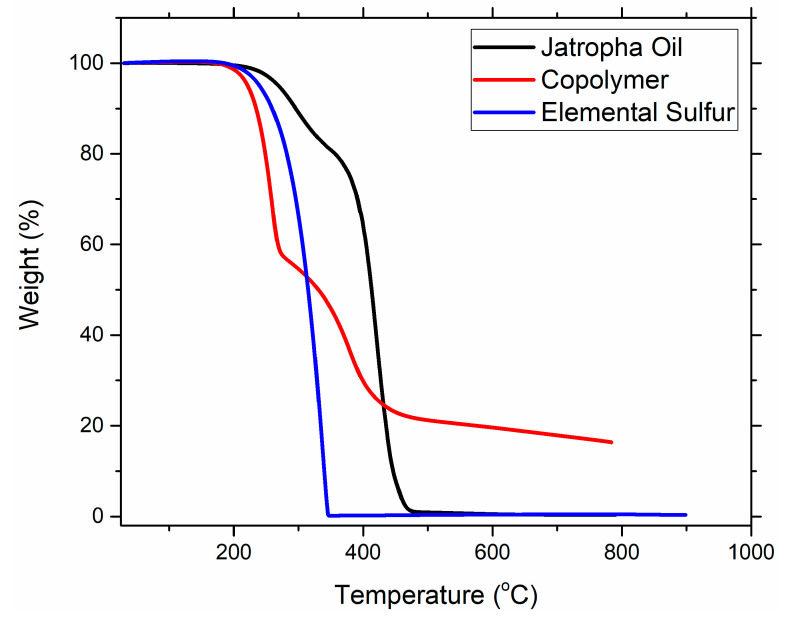
TGA thermogram of the optimized copolymer.

**Figure 6 polymers-13-04040-f006:**
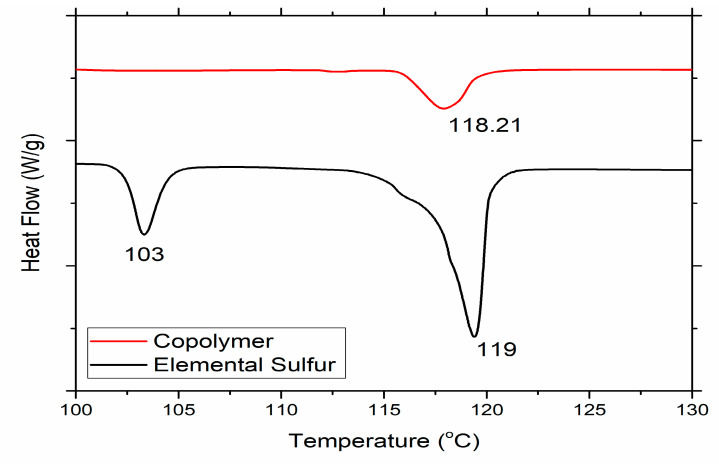
DSC thermogram of the optimized copolymer.

**Figure 7 polymers-13-04040-f007:**
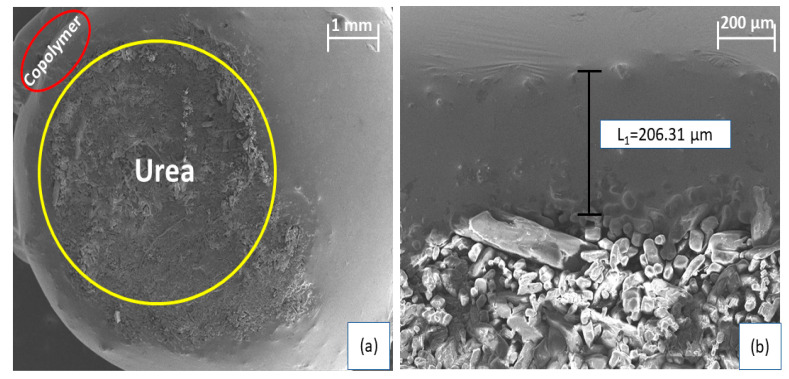
SEM images of the cross-section of the coated urea. (**a**) 1 mm. (**b**) 200 μm.

**Figure 8 polymers-13-04040-f008:**
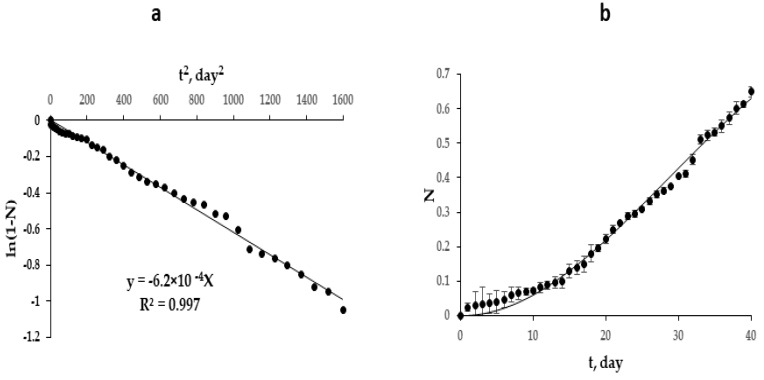
Kinetics of urea release: (**a**) linear dependence in the coordinates “ln(1 − *N*) vs. *t*^2^”; (**b**) theoretical (continuous line) and experimental (points) kinetic curves of nitrogen accumulation in water.

**Figure 9 polymers-13-04040-f009:**
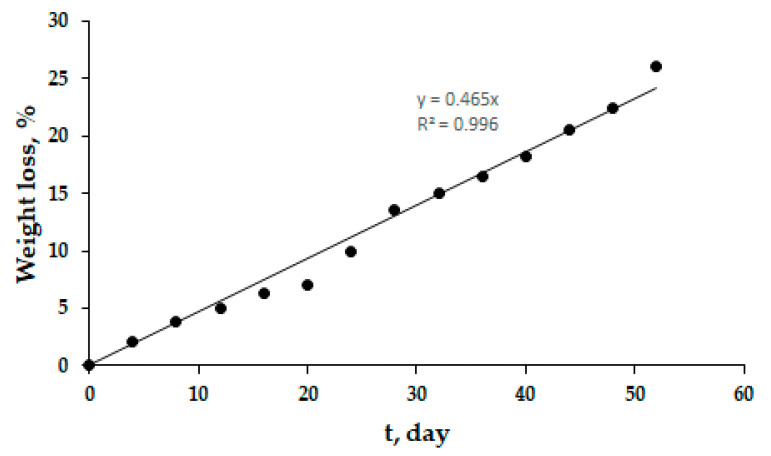
Weight loss of the terpolymer in soil.

**Table 1 polymers-13-04040-t001:** Chosen variable with their coded values.

Code	Initial Parameter	Levels				
		−1.682	−1	0	+1	+1.682
A	Sulfur content (%)	50	56.08	75	73.92	80
B	Reaction Temperature (°C)	170	173	175	177.5	185
C	Reaction Time (min)	30	48.24	75	101.8	120

**Table 2 polymers-13-04040-t002:** RSM design for three independent factors using CCD for inverse vulcanization of JO with sulfur.

	Factor 1	Factor 2	Factor 3	Response	
Run	Sulfur Content (wt%)	Temperature (°C)	Time (min)	Sulfur Conversion (%)	
				Actual	Predicted
1	65	177.5	75	65.71	63.10
2	65	177.5	75	64.23	63.10
3	56.08	182	48.24	62.23	64.26
4	65	177.5	75	62.25	63.10
5	65	177.5	30	45.2	46.08
6	65	177.5	120	69.23	66.51
7	73.91	182	48.24	37.66	37.44
8	56.08	173	48.24	70.23	68.94
9	65	170	75	57.43	58.93
10	65	177.5	75	63.35	63.10
11	56.08	182	101.8	75.83	78.07
12	80	177.5	75	22.62	22.83
13	56.08	173	101.8	77.88	79.40
14	73.91	181.96	101.8	48.63	51.24
15	73.91	173	101.8	40.2	39.46
16	73.91	173	48.24	29.9	28.96
17	65	177.5	75	64.2	63.10
18	65	185	75	68.26	64.90
19	50	177.5	75	81.06	79.01
20	65	177.5	75	58.5	63.10

**Table 3 polymers-13-04040-t003:** Full ANOVA.

Source	Sum of Squares	df	Mean Square	F-Value	*p*-Value	
**Model**	4771.56	9	530.17	67.44	<0.0001	Significant
A—Sulfur Content	3808.29	1	3808.29	484.42	<0.0001	
B—Temp	43.39	1	43.39	5.52	0.0407	
C—Time	504.42	1	504.42	64.16	<0.0001	
AB	86.36	1	86.36	10.98	0.0078	
AC	0.0006	1	0.0006	0.0001	0.9930	
BC	5.57	1	5.57	0.7085	0.4196	
A^2^	267.13	1	267.13	33.98	0.0002	
B^2^	2.52	1	2.52	0.3206	0.5837	
C^2^	83.50	1	83.50	0.9800	0.67026	
**Residual**	78.62	10	7.86			
Lack of Fit	47.39	5	9.48	1.52	0.3291	not significant
Pure Error	31.22	5	6.24			
**Cor Total**	4850.18	19				

**Table 4 polymers-13-04040-t004:** ANOVA of the Reduced Model.

Source	Sum of Squares	df	Mean Square	F-Value	*p*-Value	
**Model**	4763.46	6	793.91	119.01	<0.0001	Significant
A—Sulfur Content	3808.26	1	3808.26	570.87	<0.0001	
B—Temp	43.39	1	43.39	6.50	0.0242	
C—Time	504.40	1	504.40	75.61	<0.0001	
AB	86.32	1	86.32	12.94	0.0032	
A^2^	264.64	1	264.64	39.67	<0.0001	
**Residual**	86.72	13	6.67			
Lack of Fit	55.50	8	6.94	1.11	0.4754	not significant
Pure Error	31.22	5	6.24			
**Cor Total**	4850.18	19				

## Data Availability

The data presented in this study are available on request from the corresponding author.

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
