# Peer review of "A Degradable Inverse Vulcanized Copolymer as a Coating Material for Urea Produced under Optimized Conditions"

_polymers, 2021, doi:10.3390/polym13224040_

Round 1

Reviewer 1 Report

The manuscript polymers-1470530 entitled “ A degradable inverse vulcanized copolymer as a coating material for urea produced under optimized conditions” investigates the synthesis and optimization of a bio-based polymeric coating on the surface of sulfur.

The idea is interesting but the manuscript needs revisions before it can be recommended for publication.

Comments:

  1. The abstract should briefly state the purpose of the research, the principal results, and major conclusions. The results are missed in the current form.
  2. The reference style needs to be according to the journal instruction. For example, line 37 must be changed into [1,2] or line 41 must be changed into [3-5].
  3. The last paragraph of the Introduction section should be supported using a schematic figure. Please show the novelty of work, procedures, and notable results.
  4. The authors should be more careful about using abbreviations, while they must be defined just for the first time of usage. Although the response surface methodology (RSM) was defined it was not used in lines 98-99.
  5. Section 2.2.2 must be supported with relevant references.
  6. Tables are not in the format of the MDPI template for the manuscript. Please edit them.
  7. The quality of Figure 1 is low.
  8. The caption of Figure 6 is not correct. It should be edited to SEM images of the Cross-section of the coated urea or equivalent.

Reviewer 2 Report

The paper by Ghumman et al. reports on the use of inverse vulcanized copolymer as coatings of urea for the a gradual release of the nutrient. The topyc is important in the field of fertilizers.

In general English needs to be revised, see for example sentence on line 59-60 or 84-85.

The starting materials (sulfur and jatropha oil) are natural. However, during the production of the polymers non-natural and hazardous materials are used (Diisopropenyl benzene, diacetyl monoxime, thiosemicarbazide (TSC), Phosphoric acid, sulfuric acid and tetrahydrofuran). As in the introduction the green properties of natural polymers are reported, a not could be added about this issue. Moreover the synthesis of the materials releases H2S.

Line 27: write the long name of DIB, before the acronym

In Table 2, run 1, 2, 4, 17, 20 seem to be conducted in equal conditions. Are the author sure about that?

Section 2.2.4. Synthesis of terpolymer: how much diiso-propenyl benzene is used?

Equation 2 is not needed.

Section 3.1. 1 The models behind the statistical analysis should be explained in more detail. 

Figure 3. With the vertical scales used in the figure it is difficult to note the changes at 1660 and 804 cm-1.

Figure 4: the vertical scales should not be labelled "weight loss" with a 100% mass loss at the starting point of the measure.

Sentence on lines 332 and 333: please, write the sentence in a different way.

Line 358: explain what is DAM calorimetry method 

Round 2

Reviewer 2 Report

THe authors solved the problems I posed in my previous review.